# Antidiabetic drug utilization pattern, affordability and cost analysis in Iran from 2009 to 2021: A need to revise guideline

Saman Zartab[1,2], Hadi Abbasian[3]*

1 Pharmaceutical Sciences Research Center, Health Institute, Kermanshah University of Medical Sciences, Kermanshah, Iran, 2 Department of Pharmacology, School of Pharmacy, Kermanshah University of Medical Sciences, Kermanshah, Iran, 3 Department of Pharmacoeconomics and Pharmaceutical Administration, School of Pharmacy, Urmia University of Medical Sciences, Urmia, Iran

* abbasian.h@umsu.ac.ir

## Abstract

Diabetes Mellitus is a chronic disease that has a notable prevalence and continues to rise. It has a significant economic burden and consumes lots of financial resources of the health sector, while about half of the diabetic patients have uncontrolled diabetes. Therefore, surveilling the utilization pattern and affordability of antidiabetic drugs is vital for appropriate policymaking. It was a retrospective descriptive study that evaluated the utilization of antidiabetic medications.. The Anatomical Therapeutic Classification (ATC) was applied to categorize medicine. The Anatomical Therapeutic Classification/Defined Daily Dose (ATC/DDD) methodology was used to calculate the number of consumed Defined Daily Doses (DDDs) and the number of DDDs per 1000 inhabitants per day (DID). The Drug Utilization 90% (DU-90) method was also used to determine the most used antidiabetic drugs and compare them with the WHO essential medicine list. The affordability of antidiabetic medicine was measured by comparing the one-month consumption price with the minimum daily wage. The utilization of antidiabetic drugs increased from 22.5 DID to 63.9 DID during the study period with a 9.05 Compound Annual Growth Rate (CAGR). The overall expenditure has increased by about 18 times. Expenditure analysis showed that the cost per DDD has increased significantly over the years. The DU-90 list completely matches the WHO essential medicine list. All human insulin products were affordable, while almost all insulin analogues were unaffordable until the last year of the study period. Most novel non-insulin antidiabetic products such as empagloflozin, sitagliptin and extended-release metformin were unaffordable during the study period. The utilization of antidiabetic drugs has increased substantially. The expenditure on antidiabetic drugs has grown more rapidly than the utilization of these drugs. The number of unaffordable medicines increased during the study due to the introduction of some novel antidiabetic drugs. Therefore, it is important to revise the diabetes guidelines to promote rational drug use.

**Data availability statement:** The data underlying the results presented in the study are available from the Food and Drug Administration of The Islamic Republic of Iran. The data can be accessed at www.fda.gov.ir (email: info@fda.gov.ir). The authors confirm that the Drug statistics (Amarnameh) are published annually by the Iranian Food and Drug Administration and made available to the public. The authors also confirm that they did not have any special access privileges to the data.

**Funding:** The author(s) received no specific funding for this work.

**Competing interests:** The authors have declared that no competing interests exist.

## Introduction

Diabetes mellitus is a chronic and progressive disease. Type 1 is an autoimmune disease caused by the inability to produce insulin. Type 2 diabetes mellitus is caused by insufficient insulin production and insulin resistance [1]. If blood sugar is not controlled, the disease can lead to microvascular and macrovascular complications, affecting the nerves that regulate the functioning of various organs, such as the kidneys, eyes, and heart. It may even cause organ dysfunction and amputation [2]. In previous studies, it has been shown that the rate of death and disability caused by cardiovascular diseases in type 2 diabetic patients is 2–4 times higher than that of the non-diabetic population [3]. Complications of diabetes not only threaten the health of patients but also impose a large financial burden on the medical system. The prevalence of diabetes among adults aged 20–99 years in the world was estimated at 8.4% in 2017, which is expected to reach 9.9% by 2045. According to the data published by the International Diabetes Federation Atlas, it is estimated that approximately 463 million people with diabetes live worldwide [4,5].

The prevalence of diabetes in Iran has increased over the past years. Among the reasons for this phenomenon, we can mention the sedentary lifestyle, the prevalence of obesity, the increase in the average age and aging of the population. The prevalence of diabetes in Iran is 10.2%, and about a fifth are undiagnosed. In the meantime, about half of this patient population has uncontrolled diabetes. The blood sugar control had a negative relationship with age, which indicates that the problem of uncontrolled diabetes will deteriorate with the aging of the population. The incidence of diabetes in Iran was 1.2% over 5years [6].

Previous studies have illustrated different consumption patterns because of different health systems and management methods for controlling diabetes. For example, Carle Torre et al. studied the growth rate of antidiabetic medicine in the Netherlands and Portugal. They revealed that these countries had different absorption rates for new antidiabetic medicines and related expenditures. This study aimed to evaluate the consumption growth of antidiabetic drugs in different pharmacological groups and compare it with expenditure growth, which provides a helpful perspective for policy-makers [7].

## Method

This study is a descriptive retrospective study that used national-level data. The data was retrieved from Amarnameh, a national database that was created by gathering the data of wholesales to drugstores across the country. This database is published annually and contains the consumption data of 85 million citizens. Each medicine on the list is identified by a generic name, a unique code, dosage form, the manufacturer's or importer's name, the wholesaler's company name, ATC code, the total sales for each dosage form, and the total number of sold items.

The anatomical therapeutic classification was used to group the antidiabetic drugs: insulin and its analogues (A10A), sulfonylurea(A10BB), biguanides (A10BA), alpha-glucosidase inhibitors (A10BF), Dipeptidyl peptidase-4 inhibitors (A10BH),

thiazolinediones (A10BG), sodium glucose transport protein 2 inhibitors (A10BK) and glucagon-likepeptide-1 receptor agonists (A10BJ).

The overall and individual consumption of antidiabetic drugs over 13 years was expressed in defined daily dose terminology. The latest version of ATC/DDD guidelines was downloaded from the WHO Collaborating Center website for calculating DDDs [8]. The defined daily dose of a drug is the average maintenance dose in adults for its main indication. To calculate the number of consumed DDDs, the total quantity of each medicine, identified by its five-digit ATC code, was extracted from Amarnameh. The total amount was divided into DDD associated with that API established by the WHO Collaborating Centre for Drug Statistics. Defined daily dose per 1000 inhabitants per day was calculated to standardize the comparison between different years and countries. For calculating DID, the following formula was used: (number of DDD×1000/population ×365) [9]. The total cost and individual cost of each antidiabetic drug were extracted. The calculated expenditure includes both reimbursement from insurance and out-of-pocket payment by the patient. Then the per capita cost for each DDD was calculated. Finally, the Compound Annual Growth Rate (CAGR) was extracted [10].

According to the DU90% method, drugs that constituted 90% of the total amount of consumption were identified and their changes were investigated in different years [11,12]. The list of drugs constituting DU90% was checked with the Essential Medicines List introduced by WHO over different years.

Affordability was evaluated based on the method introduced by WHO [13]. According to this method, affordability is the number of daily minimum wages required to cover the cost of one month's use of medicine according to the treatment protocols. Therefore, a medicine is considered affordable if one's minimum daily wage covers its one-month consumption based on the treatment protocol. The minimum wage was retrieved from the annual reports of the Social Security Organization.

To eliminate the effect of inflation on the analysis, all costs were reported in US dollars, based on pharmacy retail prices that were extracted from the Amarnameh database. The data was analyzed using Microsoft Excel 2016.

## Results

The utilization of antidiabetic medications (A10) increased nearly three times from 22.5 DID to 63.9 DID during 2009–2021. The compound annual growth rate of antidiabetic drug utilization was 9.05% in this period. The consumption of insulin drugs increased from 4.1 to 11.5, indicating a compound growth rate of 8.97% between 2009 and 2021. The utilization of non-insulin antidiabetic drugs increased from 18.5 DID to 52.3 DID during 13 years, which presents a higher CAGR of consumption than insulin and its analogues (Fig 1).

In 2009, the most frequent non-insulin antidiabetic drug (A10B) that was consumed in Iran was glibenclamide (12.1 DID) followed by metformin (5.0 DID) and gliclazide (0.7 DID). There was a change in the pattern of non-insulin antidiabetic drugs during the 13 years (Table 1). In 2021, the most prevalent non-insulin antidiabetic drug was metformin (18.6DID), followed by gliclazide (9.5 DID). Glibenclamide was the third most frequent drug (6.5 DID). During this period, diabetes treatment has been diversified by introducing new medicines. For example, sitagliptin gained 10 percent of market share in six years, and Empagliflozin's market share increased to five percent in four years.

There is a similar trend to insulin drugs (A10A). In 2009, insulin isophan and regular accounted for 99% of insulin consumption. During the study period, new insulins were introduced to the market and treatment options diversified. In 2021, the most frequently consumed insulin was insulin aspart with a 40% market share. The following were insulin glargine and isophane with 29% and 15% market share, respectively (Table 1).

Overall expenditure on antidiabetic medicines has increased more than 18 times, with a 27.7% CAGR between 2009 and 2021. In 2009, the total expenditure on blood glucose-lowering drugs, excluding insulin, was $32 million. In 2021, the total cost of non-insulin antidiabetic medicines was $585 million. The insulin and its analogues had a steeper increase in total expenditure. Society paid 11 million US dollars in 2009 for insulin and its analogues, while it spent 224 million dollars on insulin and its analogues in 2021(Fig 2).

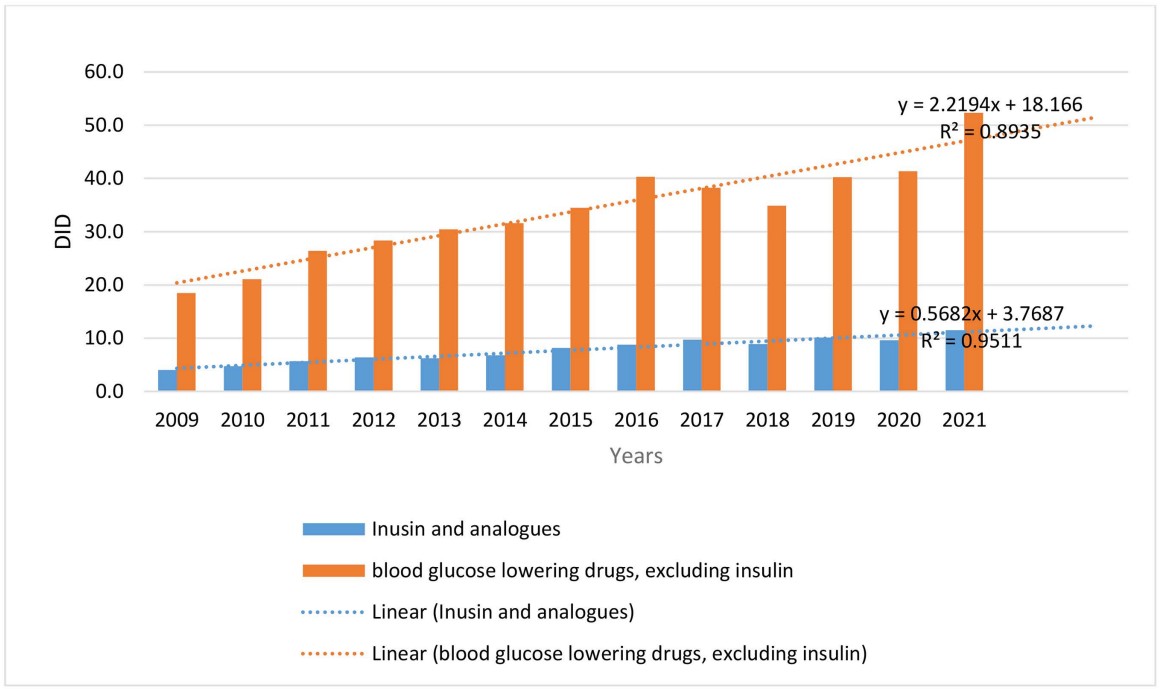

**Fig 1. Antidiabetic drug utilization.**

**Table1. Number of defined daily doses per 1000 inhabitants per day for antidiabetic medicines.**

| DID table | | 2009 | 2010 | 2011 | 2012 | 2013 | 2014 | 2015 | 2016 | 2017 | 2018 | 2019 | 2020 | 2021 |
|---|---|---|---|---|---|---|---|---|---|---|---|---|---|---|
| A10BA02 | Metformin | 5.0 | 5.6 | 8.4 | 9.3 | 12.9 | 11.3 | 11.5 | 15.1 | 12.9 | 15.1 | 16.9 | 18.6 | 18.7 |
| A10BB01 | Glibenclamide | 12.1 | 13.6 | 15.2 | 15.8 | 13.8 | 15.1 | 14.9 | 12.6 | 10.0 | 8.0 | 6.3 | 6.5 | 6.3 |
| A10BB09 | Gliclazide | 0.7 | 0.9 | 1.3 | 1.3 | 1.6 | 2.7 | 4.9 | 8.1 | 10.4 | 10.3 | 10.2 | 9.5 | 14.8 |
| A10BF01 | Acarbose | 0.1 | 0.2 | 0.2 | 0.3 | 0.3 | 0.4 | 0.4 | 0.5 | 0.5 | 0.5 | 0.5 | 0.5 | 0.5 |
| A10BG03 | Pioglitazone | 0.5 | 0.7 | 1.0 | 1.1 | 1.4 | 1.5 | 1.7 | 2.1 | 1.6 | 1.5 | 1.3 | 2.2 | 1.7 |
| A10BH01 | Sitagliptin | 0.0 | 0.0 | 0.0 | 0.0 | 0.0 | 0.0 | 0.4 | 1.0 | 1.9 | 2.9 | 3.9 | 5.1 | 5.8 |
| A10BH05 | Linagliptin | 0.0 | 0.0 | 0.0 | 0.0 | 0.0 | 0.0 | 0.0 | 0.0 | 0.0 | 0.0 | 0.5 | 0.8 | 1.0 |
| A10BX02 | Repaglinide | 0.1 | 0.2 | 0.3 | 0.5 | 0.4 | 0.6 | 0.7 | 0.8 | 0.8 | 0.6 | 0.7 | 0.8 | 0.7 |
| A10Bj02 | Liraglutide | 0.0 | 0.0 | 0.0 | 0.0 | 0.0 | 0.0 | 0.0 | 0.0 | 0.0 | 0.0 | 0.1 | 0.1 | 0.0 |
| A10BK03 | Empagliflozin | 0.0 | 0.0 | 0.0 | 0.0 | 0.0 | 0.0 | 0.0 | 0.0 | 0.0 | 0.4 | 1.0 | 1.9 | 3.0 |
| A10AB01 | Insulin Regular | 1.2 | 1.4 | 1.9 | 2.1 | 2.0 | 1.8 | 1.9 | 1.6 | 1.5 | 1.4 | 1.4 | 1.8 | 1.2 |
| A10AB05 | Insulin Aspart | 0.0 | 0.1 | 0.1 | 0.2 | 0.5 | 1.3 | 1.7 | 2.6 | 3.4 | 3.6 | 3.9 | 2.7 | 4.6 |
| A10AB06 | Insulin Glulisin | 0.0 | 0.0 | 0.0 | 0.0 | 0.0 | 0.0 | 0.0 | 0.0 | 0.1 | 0.1 | 0.3 | 0.4 | 0.4 |
| A10AE01 | Insulin Isophane | 2.9 | 3.3 | 3.6 | 4.0 | 3.5 | 3.3 | 3.4 | 2.8 | 2.2 | 1.8 | 1.4 | 2.5 | 1.8 |
| A10AE04 | Insulin Glargine | 0.0 | 0.0 | 0.0 | 0.1 | 0.2 | 0.4 | 1.2 | 1.8 | 2.2 | 2.5 | 2.9 | 2.4 | 3.3 |
| A10AE05 | Insulin Detemir | 0.0 | 0.0 | 0.0 | 0.0 | 0.0 | 0.0 | 0.0 | 0.0 | 0.1 | 0.1 | 0.2 | 0.2 | 0.3 |
| A10A | Inusin and analogues | 4.1 | 4.8 | 5.7 | 6.4 | 6.2 | 6.8 | 8.2 | 8.9 | 9.7 | 9.6 | 10.1 | 9.9 | 11.5 |
| A10B | blood glucose lowering drugs, excluding insulin | 18.5 | 21.1 | 26.4 | 28.3 | 30.4 | 31.6 | 34.5 | 40.3 | 38.3 | 39.4 | 41.3 | 46.0 | 52.3 |
| A10 | blood glucose lowering drugs | 22.6 | 25.9 | 32.1 | 34.7 | 36.7 | 38.3 | 42.7 | 49.1 | 48.0 | 48.9 | 51.4 | 55.8 | 63.9 |

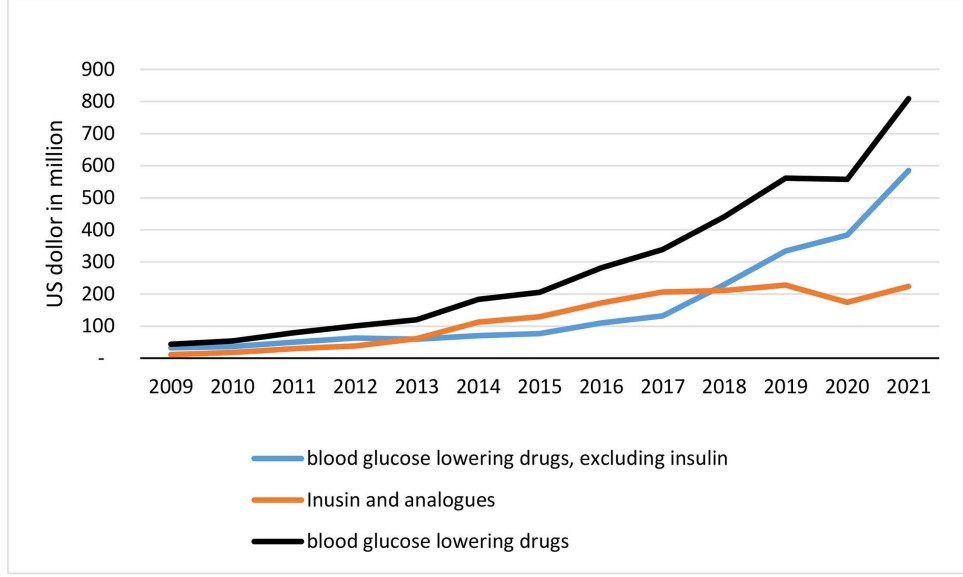

**Fig 2. Antidiabetic drugs expenditure.**

Expenditure analysis indicates that not only has consumption increased during the study period, but the cost of antidiabetic medicine has also sharply increased. The total cost per DDD rose from 0.07 $/DDD in 2009 to 0.41 $/DDD in 2021. The rise in expenditure per DDD was almost the same for insulin and non-insulin antidiabetic drugs. In 2009, the total cost of insulin per DDD was 0.1 US dollars and it has reached 0.63 $/DDD in 2021. A similar pattern can be recognized for non-insulin antidiabetic drugs. It has risen from 0.06 $/DDD to 0.36$/DDD during the study period. At the same time, GDP per capita has decreased during these days (Fig 3).

### Drug utilization 90%(DU90)

During the study period, the DU90 list for antidiabetic medicine was identified. For insulin and its analogs, in 2009, the list comprised isophane insulin and regular insulin. In 2013, insulin aspart was added to this list as the share of human insulin consumption decreased. In 2015, insulin glargine was also added to this list. By 2021, these four types of insulin accounted for 90% of national consumption, although their rankings within the list fluctuated (Table 2).

Regarding non-insulin antidiabetic drugs, in 2009, the DU90 list included glibenclamide and metformin. In 2012, with a decrease in the share of glibenclamide, gliclazide was added to this list. Following the decline in glibenclamide's market share, pioglitazone was added to this list in 2016, but it was replaced by sitagliptin the following year. This trend continued until 2020 when empagliflozin was added to this list (Table 3).

### Affordability

Among non-insulin antidiabetic drugs, metformin, glibenclamide, gliclazide, and pioglitazone were continuously affordable. Repaglinide was unaffordable for three years after its introduction to the market, but it became affordable in the rest of the study years. Unlike the regular formulation of metformin, the slow-release formulation was affordable in some years but unaffordable in others. Acarobose, which the insurance system has not covered, was unaffordable for most of the years. Sitagliptin was unaffordable in all years except the most recent one (2021) (Fig 4). Liraglutide remained unaffordable throughout the years studied, despite a price reduction following the policy of using biosimilar drugs. Among the insulin

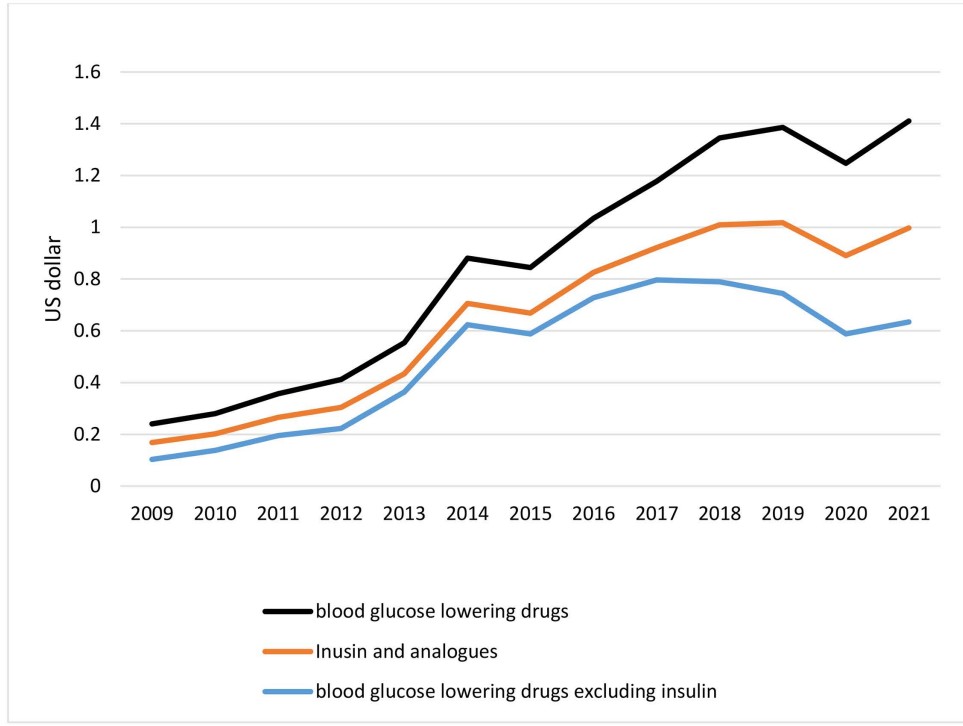

**Fig 3. Antidiabetic drugs cost per DDD.**

**Table 2. Drug Utilization-90% for insulin and its analogues.**

| 2009 | 2012 | 2016 | 2021 |
|---|---|---|---|
| Glibenclamide (66%) | Glibenclamide (55%) | Metformin(40%) | Metformin(32%) |
| Metformin (27%) | Metformin (34%) | Gliclazide(19%) | Gliclazide(25%) |
| Gliclazide (4%) | Gliclazide (5%) | Glibenclamide(30%) | Glibenclamide(21%) |
| Pioglitazone (2%) | Pioglitazone (4%) | Sitagliptin (2%) | Sitagliptin (10%) |
| Acarbose (1%) | Repaglinide (2%) | Empagliflozin(0%) | Empagliflozin(5%) |
| Repaglinide (0%) | Acarbose (1%) | Pioglitazone (5%) | Pioglitazone (3%) |
| Sitagliptin (0%) | Sitagliptin (0%) | Repaglinide (2%) | Repaglinide (2%) |
| Linagliptin (0%) | Linagliptin (0%) | Linagliptin (0%) | Linagliptin (1%) |
| Liraglutide (0%) | Liraglutide (0%) | Acarbose (1%) | Acarbose (1%) |
| Empagliflozin (0%) | Empagliflozin (0%) | Liraglutide (0%) | Liraglutide (0%) |

**Table 3. Drug Utilization-90% for non-insulin antidiabetic medicines.**

| 2009 | 2013 | 2015 | 2021 |
|---|---|---|---|
| Insulin Isophane (70%) | Insulin Isophane (56%) | Insulin Isophane (41%) | Insulin Aspart (40%) |
| Insulin Regular (29%) | Insulin Regular (33%) | Insulin Regular (23%) | Insulin Glargine (29%) |
| Insulin Aspart (1%) | Insulin Aspart (8%) | Insulin Aspart (20%) | Insulin Isophane (15%) |
| Insulin Glargine (0%) | Insulin Glargine (4%) | Insulin Glargine (15%) | Insulin Regular (10%) |
| Insulin Glulisin (0%) | Insulin Glulisin (0%) | Insulin Glulisin (0%) | Insulin Glulisin (4%) |
| Insulin Detemir (0%) | Insulin Detemir (0%) | Insulin Detemir (0%) | Insulin Detemir (2%) |

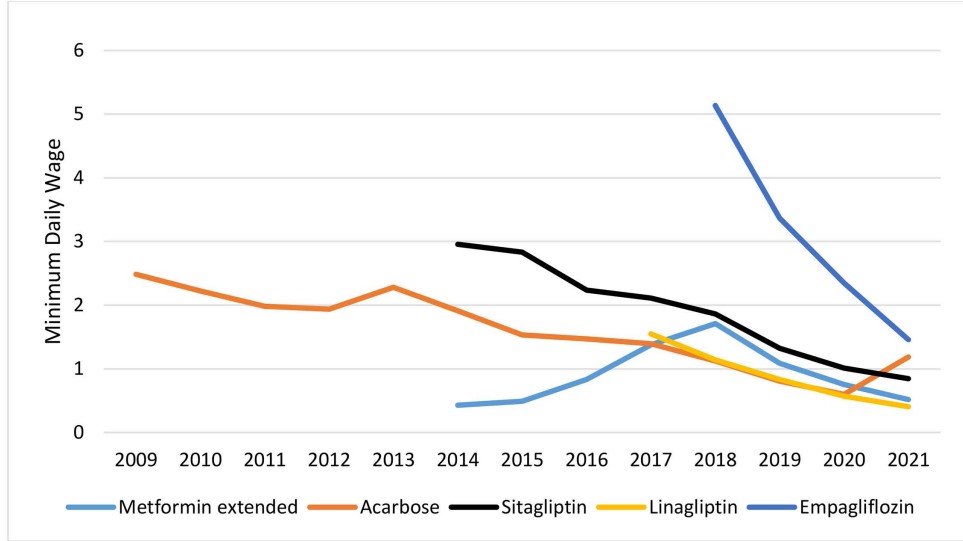

**Fig 4. Affordability of non-insulin antidiabetic medicine.**

antidiabetic medicines, human insulin products were affordable in all the years studied. Analog insulin products were previously completely unaffordable until 2020, when insulins aspart, glulisine, and glargine became affordable due to a 42% increase in minimum wages and a domestic production policy (Figs 5 and 6).

## Discussion

This study provides a long-term examination of the use of antidiabetic drugs in Iran, offering an insight into the pattern of antidiabetic drug utilization and its associated costs at the national level. Studies such as Sarayani et al., Yousefi et

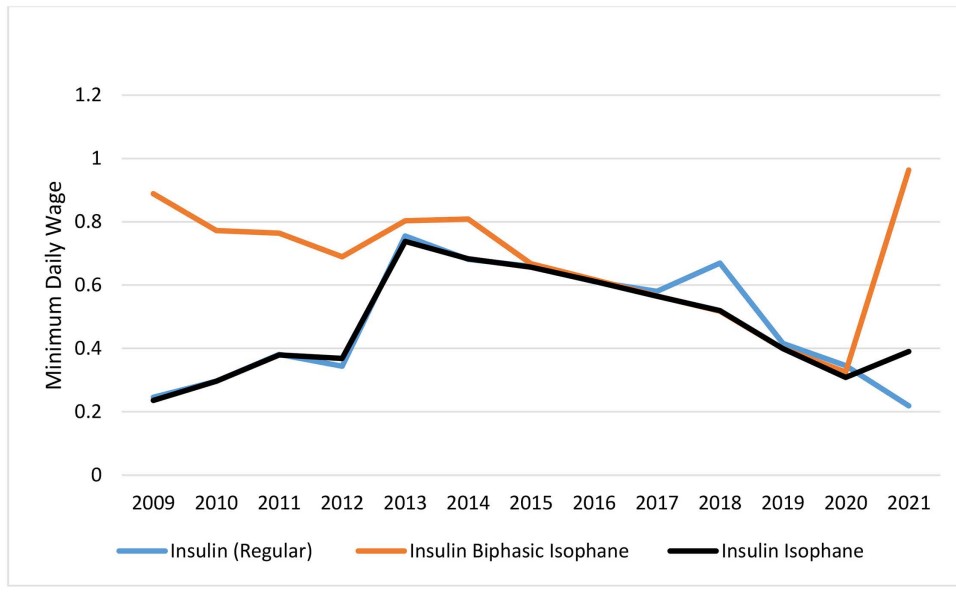

**Fig 5. Affordability of human insulin.**

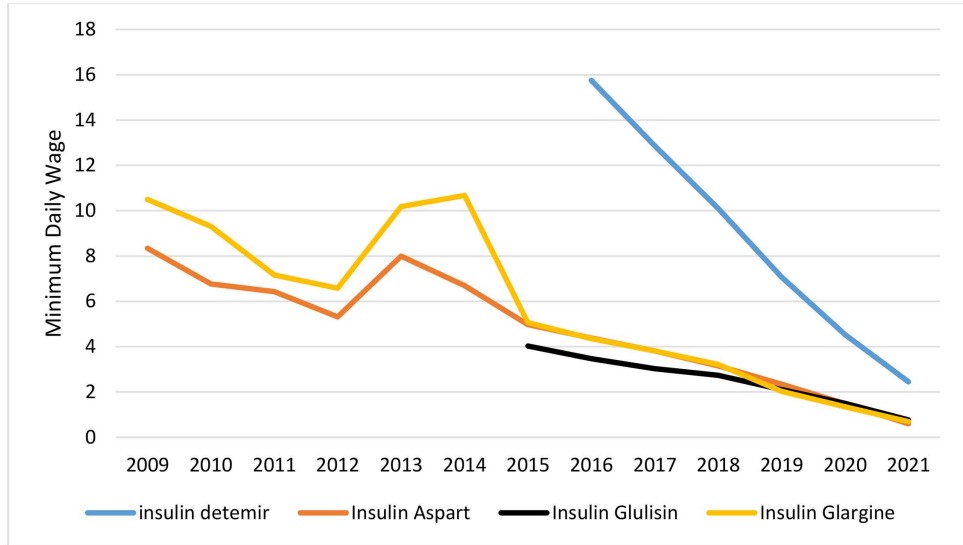

**Fig 6. Affordability of insulin analogues.**

al., and Kheirandish et al. have investigated the consumption of antidiabetic drugs, which differ from the present study in terms of time period and details of cost analysis. [9,14,15] During the studied time, new pharmacological groups were introduced to the market and changed the market share of antidiabetic drugs. However, biguanides and sulfonylureas still had the largest market share. The utilization of metformin has grown over the years, as observed in many other studies, which is in accordance with international guidelines, such as those from the American Diabetes Association (ADA) and the European Association for the Study of Diabetes (EASD) [16–19]. These guidelines recommend metformin as the first-line treatment at the initial diagnosis of diabetes, because metformin has a low risk of hypoglycemia and can help reduce weight.

Additionally, it decreases the total mortality risk compared to sulfonylureas and insulin [20,21]. Although sulfonylureas remain the second most commonly used oral antidiabetic drugs, during this period, they lost market share to new pharmacological groups introduced to the diabetic market. Afterwards, some guidelines, such as those from NICE, stated that the combined use of metformin and glibenclamide increases the mortality rate due to hypoglycemia; the use of sulfonylureas, especially glibenclamide, has decreased [16,22]. Some studies have also stated that the consumption of sulfonylureas increases all-cause mortality and mortality related to cardiovascular events [23,24].

During the study period, the consumption of thiazolidinediones did not increase due to the publication of studies showing an increased risk of developing heart failure, weight gain, and bone fractures [25–28]. The consumption of alpha-glucosidase inhibitors did not grow due to the lack of efficiency in reducing glycated hemoglobin (HbA1c) and gastrointestinal complications [29].

Among the new pharmacological groups that have recently been introduced to the market, Dipeptidyl peptidase 4 (DPP-4) inhibitors and Sodium-Glucose Transport Protein 2 (SGLT2) Inhibitors have experienced a very rapid consumption growth. However, their application is not the first line of treatment, and the number of patients eligible for these drugs is limited. Some international guidelines recommend the addition of DPP-4 and SGLT-2 drugs after metformin monotherapy failure [17,30]. Therefore, many formulations of these drugs, single or in combination with metformin, were marketed during the past years. Glucagon-like peptide-1 (GLP-1) agonists are another newly introduced pharmacological group. At the time of the study, liraglutide was the only one on the market; however, due to its high price, it could not gain a significant share of the diabetic market.

The newly introduced drugs have a considerable beneficial effect on blood sugar control. In addition, SGLT-2 and GLP-1RA drugs have a positive effect on weight loss and cardiovascular outcomes; however, their long-term effects on mortality and safety have not been fully established [31,32]. Currently, after metformin monotherapy failure, there is no consensus on the second line of treatment. After adding the new drugs to the national drug list, physicians prescribe one of them based on the patient's condition.

During the study period, insulin consumption increased from 4.0 DID to 11.5 DID, which is partly due to the rise in the number of newly diagnosed patients in the community and partly due to the increase in the number of patients with advanced stages of the disease. The Total expenditure on insulin increased from $11.3 million to $223.9 million between 2009 and 2021. Insulin consumption has grown about three times, and the growth rate of the insulin budget was approximately 20 times, which is the result of the introduction of new insulin analogs to the market.

In the early years of the study, human insulins were predominant on the market. However, during the study period, analog insulins, which offered more convenient timing and usage, replaced human insulins. For instance, long-acting insulin analogs such as glargine insulin and detemir insulin, which have fewer side effects on weight gain and nocturnal hypoglycemia, have gradually been replacing NPH insulin [33,34]. Short-acting insulin analogs, such as glulisine and aspart insulin, which can be used immediately before meals due to their improved pharmacokinetic properties, have replaced regular insulin [35]. This replacement intensified after insulin analogues were covered by insurance and the price and the patient's out-of-pocket payment decreased. Although insulin analogs have improved properties, they are more expensive and resource-intensive. Therefore, the cost per DDD for insulin has increased six times during these years.

Even though newly introduced antidiabetic medications were initially not covered by insurance, their acceptance rate was high. One reason is the medical community's passion for the latest treatments. Another issue is the lack of national guidelines, which leads physicians to adhere to guidelines from developed countries. Therefore, the lack of attention to the budget and economics of treatment has caused the cost per DDD of diabetes treatment to increase six fold over the last 13 years.

It was predictable that due to the increase in the population, the aging and spreading of sedentary lifestyles, the consumption of antidiabetic drugs would increase. In comparison, it was remarkable that the cost of antidiabetic drugs has increased more than their consumption. Therefore, the number of unaffordable antidiabetic drugs has increased. In this situation, completing and updating national guidelines, pursuing policies to adhere to these guidelines, and monitoring their implementation can help prevent this trend from continuing.

Due to domestic production and pricing policies, human insulins were affordable throughout the study period. Insulin Analogs were unaffordable, but due to negotiations with insurance, they moved towards affordability until 2021, when they became affordable as a result of domestic production. Despite domestic production and a sharp price drop, liraglutide remained unaffordable. Its affordability increased from a 25 working-day payment to a 5.5 working-day payment for consuming 30 DDD.

Most of the oral antidiabetic drugs were affordable during the study period. Meanwhile, acarbose and metformin extended-release were affordable in some years and unaffordable in others. Although sitagliptin and empagloflozin were domestically produced from the time of their introduction to the market, they remained unaffordable during the study period. Lingliptin was initially unaffordable, but it became affordable in 2019.

A possible explanation for the observed trend is that traditionally, insurance organizations like the Social Security Organization had a dominant role in the pharmaceutical supply chain by acquiring pharmaceutical companies and following the strategy of producing low-cost drugs to reduce their expenses. Gradually, during the study period, the market share of pharmaceutical companies owned by insurance organizations has decreased, while private companies have gained a significant market share. Now, more than 50% of the drugs used in Iran are produced by private companies. In this situation, it is expected that the number of unaffordable drugs will increase in the near future, because private pharmaceutical companies, unlike those possessed by insurance organizations, do not have an incentive to reduce drug prices. For instance,

despite the generic production of sitagliptin and empagliflozin by private companies for several years, these drugs remain unaffordable due to aggressive marketing and branding strategies.

In Iran, the Pricing Commission of the Food and Drug Administration is authorized for drug pricing. Historically, several policies have been implemented to enhance the accessibility and affordability of drugs. First, to increase access to medicine, high prices are permitted for newly approved imported and domestically produced drugs. As general inflation increases, companies' expenses and wages rise, but annual price increases are not permitted. In spite of this, companies are not interested in leaving the market after incurring marketing expenses and gaining market share. In the next step, in order to gain insurance coverage that leads to an increase in sales, insurance organizations negotiate with pharmaceutical companies to reduce their prices. Ultimately, insurance organizations that own a significant number of pharmaceutical manufacturing plants follow a strategy of producing low-cost drugs and stabilizing drug prices. However, with the declining market share of insurance-based pharmaceutical companies, achieving affordable prices through this strategy will become challenging. However, it is hoped that in the presence of private companies and higher drug prices, stricter oversight by the Food and Drug Administration will improve the quality of domestically produced drugs.

One of the strengths of this study is its comprehensive coverage of the entire Iranian population. The study encompasses all consumption data for both outpatients and inpatients, including all sales of diabetic medications, regardless of whether they are covered by insurance or not. The use of sales data from distributors to pharmacies is considered one of the study's weak points because a portion of the purchased medicines may not be consumed by patients or may even expire in pharmacies. Another limitation of the study is the inability to differentiate between patients with type 1 and type 2 diabetes who use insulin. Finally, although the definition of ability to pay is based on the official minimum wage, many people work at prices lower than the official minimum wage.

While this study has evaluated the consumption of antidiabetic drugs and their associated expenditure, it is worth mentioning that for effective policymaking, the various costs related to the complications of inadequate diabetes management should also be considered.

In conclusion, the utilization of antidiabetic medicine has increased notably in Iran. The growth rate of expenditure exceeds the utilization of antidiabetic medicine. Due to the release of several novel antidiabetic medications, the number of unaffordable drugs increased during the study period, although they are moving toward affordability.

## Acknowledgments

We would like to appreciate IFDA for their declaration of information

## Author contributions

**Conceptualization:** Saman Zartab, Hadi Abbasian.

**Data curation:** Saman Zartab, Hadi Abbasian.

**Formal analysis:** Saman Zartab, Hadi Abbasian.

**Funding acquisition:** Saman Zartab, Hadi Abbasian.

**Investigation:** Saman Zartab, Hadi Abbasian.

**Methodology:** Saman Zartab, Hadi Abbasian.

**Project administration:** Hadi Abbasian.

**Resources:** Saman Zartab, Hadi Abbasian.

**Software:** Saman Zartab, Hadi Abbasian.

**Supervision:** Saman Zartab, Hadi Abbasian.

**Validation:** Hadi Abbasian.

**Visualization:** Saman Zartab, Hadi Abbasian.

**Writing – original draft:** Saman Zartab.

**Writing – review & editing:** Hadi Abbasian.

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
