## [Decision Letter · Decision Letter 0]

24 Apr 2025

Dear Dr. Abbasian,

Thank you for submitting your manuscript to PLOS ONE. After careful consideration, we feel that it has merit but does not fully meet PLOS ONE’s publication criteria as it currently stands. Therefore, we invite you to submit a revised version of the manuscript that addresses the points raised during the review process.

The manuscript has been evaluated by two reviewers, and their comments are available below.



We look forward to receiving your revised manuscript.

Kind regards,

Steve Zimmerman, PhD

Senior Editor, PLOS One

Journal Requirements:

Reviewers' comments:

Reviewer's Responses to Questions

**Comments to the Author**

1. Is the manuscript technically sound, and do the data support the conclusions?

Reviewer #1: Partly

Reviewer #2: Partly

2. Has the statistical analysis been performed appropriately and rigorously?

Reviewer #1: N/A

Reviewer #2: I Don't Know

3. Have the authors made all data underlying the findings in their manuscript fully available?

Reviewer #1: Yes

Reviewer #2: Yes

4. Is the manuscript presented in an intelligible fashion and written in standard English?

Reviewer #1: Yes

Reviewer #2: Yes

Reviewer #1: This study presents valuable insights into the utilization and cost trends of antidiabetic medications in Iran over a 13-year period. The manuscript addresses important topics such as affordability and market dynamics; however, there are several areas that require improvement, as outlined below:

1. Use of Abbreviations: The manuscript does not consistently introduce abbreviations before their first use. Several abbreviations, such as DDD, ATC, DID, DU, GDP, ADA, and ESAD, … are used without prior definition, which may hinder comprehension—particularly in the abstract. It is strongly recommended that the authors carefully review the whole manuscript and ensure that all abbreviations are properly defined upon first mention.

2. Comparison with Existing Literature: The study utilizes public data from Amarnameh; however, data from Amarnameh on trends in blood glucose-lowering medications have been previously analyzed and reported in other published works and the manuscript does not acknowledge these prior studies, nor does it compare its findings to them. For instance, the following study provides relevant insights:

Kheirandish M, Varahrami V, Kebriaeezade A, Cheraghali AM. Impact of economic sanctions on access to noncommunicable diseases medicines in the Islamic Republic of Iran. East Mediterr Health J. 2018 Apr 5;24(1):42-51. PMID: 29658620.

I recommend the authors to clarify how their study differs from previous works, whether it provides additional insights, updates the findings, or introduces a new analytical approach.

3. Consideration of Bias and Confounding Factors: Given the retrospective nature of the study, it is important to address potential sources of bias and confounding factors. For example, the assumption that minimum wage accurately reflects purchasing power may not be entirely valid, as income disparities exist within the population. Additionally, changes in healthcare policies, pricing regulations, or diagnostic practices may have influenced the observed trends. While some policy changes are briefly mentioned (e.g., line 196: Liraglutide was unaffordable in all the years studied, despite the price reduction following the policy of using biosimilar drugs), a more detailed discussion of their details and impact on the observed trends would enhance the manuscript.

4. Title and Recommendations: The title includes the phrase "a need to revise guidelines”, yet the manuscript does not provide clear, actionable recommendations based on the findings. While the study highlights the economic burden of antidiabetic drugs, it would benefit from a discussion on potential policy adjustments or strategies to address the observed trends.

5. Speculative Statements: Some sections of the discussion contain statements that appear speculative or opinion-based rather than being fully supported by data or citations of the previous literature. Specifically, lines 247–248 and 294–322 should be revised to either include supporting evidence from the literature or be explicitly framed as author’s hypotheses to avoid misinterpretation.

6. Lack of Statistical Analysis: A key limitation of the study is the absence of statistical analysis. While the manuscript presents descriptive statistics (e.g., trends over time, market share changes), it does not employ inferential statistical methods to assess the significance of these trends. For instance, regression analysis could be used to evaluate the relationship between drug costs and utilization over time or to compare the affordability of different drug classes. Incorporating statistical analysis would strengthen the validity of the conclusions drawn.

Reviewer #2: Overall, this is a well-structured and well-written manuscript. I have a few comments for some revision. The introduction section is relevant; however, it does not provide a clear overview of the background and rationale for the study. The study's objectives and procedures must be added and presented logically, making it easy for the reader to follow.

You need to include global, regional, and then Iranian diabetes prevalence.

In the abstract, lines 31-32, first write the full meaning of DID abbreviation.

Line 95 (please use × sing instead of *) DDD*1000/, please use DDD×1000

Line 144, for Table heading years, is broken down; you need to adjust the table size or change it to a smaller font size.

Line 170, the full meaning of DU abbreviation.

Figure 2, for the vertical axis heading, please add "US dollar in million"

Figure 4, in the horizontal axis, years are not readable; please adjust the figure size or make the year's font smaller to make it readable. The same also applies to figure 5

You may add the generalizability of this study's findings before the conclusion.

For the method section, you may add a few more statistics such as 95% confidence intervals for Insusin and analogues and blood glucose lowering drugs. For Figures 2 and 3, you should check to use the Chi-square test or any other type of test.

If you want to compare the mean outcome (antidiabetic medicine expenditure) across the three medicines, you can use statistical test and suggested test5 is one-way ANOVA.

Please also check the English and spelling throughout the paper.

Please add Iran in the conclusion.

**Do you want your identity to be public for this peer review?** For information about this choice, including consent withdrawal, please see our Privacy Policy

Reviewer #1: No

Reviewer #2: **Yes: ** Montasir Ahmed

---

## [Author Response · Author response to Decision Letter 1]

29 Jun 2025

Reviewer #1: This study presents valuable insights into the utilization and cost trends of antidiabetic medications in Iran over a 13-year period. The manuscript addresses important topics such as affordability and market dynamics; however, there are several areas that require improvement, as outlined below:

1. Use of Abbreviations: The manuscript does not consistently introduce abbreviations before their first use. Several abbreviations, such as DDD, ATC, DID, DU, GDP, ADA, and ESAD, … are used without prior definition, which may hinder comprehension—particularly in the abstract. It is strongly recommended that the authors carefully review the whole manuscript and ensure that all abbreviations are properly defined upon first mention.

Response to reviewer1 #1:

Thanks to the careful attention of the esteemed reviewer, the abbreviations in the text have been corrected.

2. Comparison with Existing Literature: The study utilizes public data from Amarnameh; however, data from Amarnameh on trends in blood glucose-lowering medications have been previously analyzed and reported in other published works and the manuscript does not acknowledge these prior studies, nor does it compare its findings to them. For instance, the following study provides relevant insights:

Kheirandish M, Varahrami V, Kebriaeezade A, Cheraghali AM. Impact of economic sanctions on access to noncommunicable diseases medicines in the Islamic Republic of Iran. East Mediterr Health J. 2018 Apr 5;24(1):42-51. PMID: 29658620.

I recommend the authors to clarify how their study differs from previous works, whether it provides additional insights, updates the findings, or introduces a new analytical approach.

Response to reviewer1 #2:

Thanks to the reviewer’s valuable comments, the requested material was added and referenced.

3. Consideration of Bias and Confounding Factors: Given the retrospective nature of the study, it is important to address potential sources of bias and confounding factors. For example, the assumption that minimum wage accurately reflects purchasing power may not be entirely valid, as income disparities exist within the population. Additionally, changes in healthcare policies, pricing regulations, or diagnostic practices may have influenced the observed trends. While some policy changes are briefly mentioned (e.g., line 196: Liraglutide was unaffordable in all the years studied, despite the price reduction following the policy of using biosimilar drugs), a more detailed discussion of their details and impact on the observed trends would enhance the manuscript.

Response to reviewer1 #3:

Thanks for your suggestions. According to the World Health Organization, “affordability” depends on the purchasing power and price of the product, and many people in Iranian society work below the declared minimum wage, which has been added to the text.

Unfortunately, there has been no change in the diagnostic methods during the studied period in Iran, and the Cost Plus method has been used for drug pricing in Iran for a long time. The only significant policy change in this matter is the growth of the private sector and their market share, which was mentioned in the text.

4. Title and Recommendations: The title includes the phrase "a need to revise guidelines”, yet the manuscript does not provide clear, actionable recommendations based on the findings. While the study highlights the economic burden of antidiabetic drugs, it would benefit from a discussion on potential policy adjustments or strategies to address the observed trends.

Response to reviewer1 #4:

Added with thanks.

5. Speculative Statements: Some sections of the discussion contain statements that appear speculative or opinion-based rather than being fully supported by data or citations of the previous literature. Specifically, lines 247–248 and 294–322 should be revised to either include supporting evidence from the literature or be explicitly framed as author’s hypotheses to avoid misinterpretation.

Response to reviewer1 #5:

Thanks, corrected in the text.

6. Lack of Statistical Analysis: A key limitation of the study is the absence of statistical analysis. While the manuscript presents descriptive statistics (e.g., trends over time, market share changes), it does not employ inferential statistical methods to assess the significance of these trends. For instance, regression analysis could be used to evaluate the relationship between drug costs and utilization over time or to compare the affordability of different drug classes. Incorporating statistical analysis would strengthen the validity of the conclusions drawn.

Response to reviewer1 #6:

Time series regression equations were added to examine the trend of increasing consumption, and Figure 1 was modified.

Reviewer #2: Overall, this is a well-structured and well-written manuscript. I have a few comments for some revision. The introduction section is relevant; however, it does not provide a clear overview of the background and rationale for the study. The study's objectives and procedures must be added and presented logically, making it easy for the reader to follow.

You need to include global, regional, and then Iranian diabetes prevalence.

Thanks for your attention, it has been corrected in the text.

In the abstract, lines 31-32, first write the full meaning of DID abbreviation.

Thanks for your attention, it has been corrected in the text.

Line 95 (please use × sing instead of *) DDD*1000/, please use DDD×1000

Thanks for your attention, it has been corrected in the text.

Line 144, for Table heading years, is broken down; you need to adjust the table size or change it to a smaller font size.

The table has been adjusted.

Line 170, the full meaning of DU abbreviation.

The abbreviation was changed to the whole word, and the whole word was added to the abstract.

Figure 2, for the vertical axis heading, please add "US dollar in million"

Thanks for your attention, it has been corrected in the text.

Figure 4, in the horizontal axis, years are not readable; please adjust the figure size or make the year's font smaller to make it readable. The same also applies to figure 5

You may add the generalizability of this study's findings before the conclusion.

Thanks for your attention, it has been corrected in the text.

For the method section, you may add a few more statistics such as 95% confidence intervals for Insusin and analogues and blood glucose lowering drugs. For Figures 2 and 3, you should check to use the Chi-square test or any other type of test.

If you want to compare the mean outcome (antidiabetic medicine expenditure) across the three medicines, you can use statistical test and suggested test5 is one-way ANOVA.

Given that this study uses a national database and is not a hospital-level study, the data is for the total population and generalization from the sample to the population has not occurred. Therefore, statistical tests that analyze sample data to express the probability of accuracy of the results in the statistical population are not applicable to it. So time series regression equations were added to examine the trend of increasing consumption, and Figure 1 was modified.

Please also check the English and spelling throughout the paper.

Thanks, the corrections have been made.

Please add Iran in the conclusion.

Thanks, your point has been added and highlighted.

---

## [Decision Letter · Decision Letter 1]

13 Aug 2025

Dear Dr. Abbasian,

Thank you for submitting your manuscript to PLOS ONE. After careful consideration, we feel that it has merit but does not fully meet PLOS ONE’s publication criteria as it currently stands. Therefore, we invite you to submit a revised version of the manuscript that addresses the points raised during the review process.

**ACADEMIC EDITOR: **

We look forward to receiving your revised manuscript.

Kind regards,

Eyob Alemayehu Gebreyohannes, PhD

Academic Editor

PLOS ONE

Journal Requirements:

Reviewers' comments:

Reviewer's Responses to Questions

**Comments to the Author**

Reviewer #1: All comments have been addressed

Reviewer #2: All comments have been addressed

2. Is the manuscript technically sound, and do the data support the conclusions?

Reviewer #1: Yes

Reviewer #2: Yes

3. Has the statistical analysis been performed appropriately and rigorously?

Reviewer #1: Yes

Reviewer #2: Yes

4. Have the authors made all data underlying the findings in their manuscript fully available?

Reviewer #1: Yes

Reviewer #2: Yes

5. Is the manuscript presented in an intelligible fashion and written in standard English?

Reviewer #1: Yes

Reviewer #2: Yes

Reviewer #1: I would like to thank the authors for their thoughtful revision and comprehensive responses to the comments.

Reviewer #2: Thank you for the revised version. It appears that the authors have addressed all of my comments, and the manuscript reads well. I am pleased to recommend acceptance.

However, I would like to note a few additional points for improvement before final submission:

Language and Style

While the study contains valuable data, the manuscript still suffers from grammatical errors, which may hinder reader comprehension. A thorough professional English language edit is strongly recommended.

Figures and Tables

Readability: The current use of light ash colour in Figures 1–6 makes key elements difficult to read. Please use black for improved visibility. For example, in Figure 5, the "Minimum daily wage" lines and the horizontal axis years would be more legible if displayed in black.

Figure Titles: Please remove titles from inside the figures. For instance, Figure 2 is titled “Antidiabetic drugs expenditure” in the caption but also contains the title “Antidiabetic medicine expenditure” within the figure itself. This duplication detracts from the professional appearance. The same applies to other figures—retain the title only in the caption, not within the figure.

Addressing these issues will significantly improve the presentation and clarity of your work.

**Do you want your identity to be public for this peer review?** For information about this choice, including consent withdrawal, please see our Privacy Policy

Reviewer #1: No

Reviewer #2: **Yes: ** Montasir Ahmed

---

## [Author Response · Author response to Decision Letter 2]

3 Sep 2025

Reviewer 2:

Language and Style

While the study contains valuable data, the manuscript still suffers from grammatical errors, which may hinder reader comprehension. A thorough professional English language edit is strongly recommended.

Thank you for your suggestion, the article was edited and improved in terms of English.

Figures and Tables

Readability: The current use of light ash colour in Figures 1–6 makes key elements difficult to read. Please use black for improved visibility. For example, in Figure 5, the "Minimum daily wage" lines and the horizontal axis years would be more legible if displayed in black.

According the suggestion of reviewer 2 and editor the ash color changed to black in all part of the figures so visibility od them enhanced.

Figure Titles: Please remove titles from inside the figures. For instance, Figure 2 is titled “Antidiabetic drugs expenditure” in the caption but also contains the title “Antidiabetic medicine expenditure” within the figure itself. This duplication detracts from the professional appearance. The same applies to other figures—retain the title only in the caption, not within the figure.

Titles removed from inside the figures.

ACADEMIC EDITOR:

While the manuscript has merit, the language must undergo a thorough professional edit to address grammatical errors, awkward phrasing, and inconsistent tense usage, ensuring clarity and fluency throughout.

In addition, the presentation of figures must be improved by enhancing visibility and removing embedded titles to avoid duplication with captions.

Any changes made must be indicated as tracked changes.

These refinements are essential to enhance readability and professional presentation. Please note that failure to fully address these concerns in the revised submission may result in rejection of the manuscript.

All corrections and improvements to the article's text have been made based on your suggestions.

---

## [Editor Report · Decision Letter 2]

10 Sep 2025

Anti-diabetic drug utilization pattern, affordability and cost analysis in Iran from 2009 to 2021; a need to revise guideline

PONE-D-24-56063R2

Dear Dr. Abbasian,

We’re pleased to inform you that your manuscript has been judged scientifically suitable for publication and will be formally accepted for publication once it meets all outstanding technical requirements.

Kind regards,

Eyob Alemayehu Gebreyohannes, PhD

Academic Editor

PLOS ONE

---

## [Editor Report · Acceptance letter]

PONE-D-24-56063R2

PLOS ONE

Dear Dr. Abbasian,

I'm pleased to inform you that your manuscript has been deemed suitable for publication in PLOS ONE. Congratulations! Your manuscript is now being handed over to our production team.

Kind regards,

on behalf of

Dr. Eyob Alemayehu Gebreyohannes

Academic Editor

PLOS ONE